# Quantum Dot Reflective Semiconductor Optical Amplifiers: Optical Pumping Compared with Electrical Pumping

**DOI:** 10.3390/nano12132143

**Published:** 2022-06-22

**Authors:** Farshad Serat Nahaei, Ali Rostami, Peyman Mirtaheri

**Affiliations:** 1Photonics and Nanocrystals Research Lab. (PNRL), Faculty of Electrical and Computer Engineering, University of Tabriz, Tabriz 5166/614761, Iran; farshad.serat98@ms.tabrizu.ac.ir; 2SP-EPT Lab., ASEPE Company, Industrial Park of Advanced Technologies, Tabriz 5166/614761, Iran; 3Department of Mechanical, Electronics and Chemical Engineering, OsloMet—Oslo Metropolitan University, 0167 Oslo, Norway

**Keywords:** reflective semiconductor optical amplifier, QDs, optical pumping, electrical pumping

## Abstract

A comprehensive study has been conducted on quantum dot reflective semiconductor optical amplifiers (QD-RSOAs) with optical pumps (OPs). Moreover, few studies have been completed on OP-based QD-RSOAs. A comparison is made between them and QD-RSOAs with electrical pumps (EPs) in this study. It is shown that the dynamical properties of the device can significantly develop in the optical pumping version. The optical properties are studied for both methods. Moreover, by solving the coupled differential rate and signal propagation equations, the operation of the device in the pulse mode is investigated. Finally, it is proven that OP QD-RSOAs can perform significantly better in applications such as fast all-optical signal processing and wavelength division multiplexing in passive optical networks.

## 1. Introduction

Due to quantum dot semiconductor optical amplifiers’ (QD-SOAs) promising features in optical applications such as signal processing and high-bit-rate optical switching, they have always been a subject of interest [1,2,3]. Compared with quantum-well and bulk SOAs, QD-SOAs are more advantageous due to the existence of a gap between the wetting layer (WL) and the QD levels. In addition, lower frequency chirps, shorter carrier relaxation times and a lower gain saturation have been reported in these SOAs due to the lower cross-sections in carrier–photon interactions [4]. Because of the high speed of SOAs, all current noises may be regenerated in the device’s output in a wide frequency range. Therefore, electronic devices attached to SOAs must be protected against source instabilities, electromagnetic interferences, etc. [5]. In QD-SOAs with electrical pumping, electrodes with a uniform surface across the waveguide are used. As a result, a consistent injection current is produced along the length of the SOA. However, because of the changes in optical intensity across the length of the SOA, the carrier concentration is not entirely uniform. In addition, not every point across the length of the device is affected uniformly via the injected current. For instance, on the entrance facet of the SOA, the effect of the injection current on saturation power is of lower concern than the effect of the injection current on the noise figure. However, on the output facet of the SOA, the impact of the injected current on the saturation is more important [6]. Therefore, to control the injection current, complicated methods have been proposed. By utilizing optimized currents, one can reduce crosstalk, improve gain linearity, and derive better efficiency [6]. Optical pumping is an attractive alternative to electrical pumping due to several reasons. First, the generated carriers in the QDs are not needed to be transported through the structure. Hence, a uniform carrier distribution is obtained through many QDs. Second, the device is not required to be doped, minimizing optical losses and simplifying fabrication. Third, the pump laser and device can be integrated into an identical system. Vertical cavity semiconductor optical amplifiers [7] and vertical cavity surface-emitting lasers [8] are good examples of optically pumped devices.

Because of the rapid growth of internet traffic, there is a growing trend toward wavelength division multiplexed passive optical networks (WDM-PONs). Passive optical networks are broadly used in fiber-to-the-home setups. In addition, more optical network unit bandwidth is acquired using optical amplification. Nowadays, reflective semiconductor optical amplifiers, used as colorless modulators, have engaged significant attention in wavelength-separated optical network units [9,10]. They are widely used in wideband communication systems [3,11]. QD-RSOAs have all the advantages of QD-SOAs. These include negligible crosstalk, broad bandwidth, fast dynamics in gain saturation, and large modulation bandwidth [12,13,14,15]. These features, combined with their reflective facet, make them an exciting device to investigate [16]. However, they need complicated modeling due to the effect of counter-propagating pulses on carrier dynamics.

In this paper, a comprehensive study is presented on the small-signal gains, the dynamics of carrier densities, and the saturation attributes of optically pumped (OP) QD-SOAs in their reflective configuration (QD-RSOAs). To refill the excited state (ES) in OP-RSOAs, an optical pump must be utilized. This process is achieved via the absorption of carriers into the ES. To confirm the benefits of optical pumping, a comparison is conducted between optically and electrically pumped (EP) QD-RSOAs. They have greater operation speed and a better gain recovery process. However, due to the phonon bottleneck phenomenon, carrier relaxation from the WL (wetting layer) into the ES or the GS (ground state) is challenging [17,18]. Maximum output densities, optical transparency power, and transparency currents are discussed to compare EP and OP QD-RSOAs. An analytical solution is obtained to derive the desired results for the continuous wave (CW) operation mode at a steady state. In addition, numerically solved rate and signal propagation equations are used to derive the pulse mode operation. Finally, the effect of the reflection on the output signal power is studied.

## 2. Concept and Modelling

### 2.1. Rate and Signal Propagation Equations

In the suggested model, QDs come with a two-dimensional wetting layer (WL). In addition, they are under quasi-equilibrium. As depicted in Figure 1, two energy states are confined in valence and conduction bands. Optical properties of the QD-RSOAs with optical or electrical pumping are defined by the rate equations [2,19,20], but reflections of the end facet must be included. They represent the dynamics of the WL and the energy states listed below:(1)$$\frac{\partial N_{W}(z,t)}{\partial t}=\frac{I}{qV}-\frac{N_{W}(1-h)}{\tau_{we}}+\frac{N_{Q}h}{\tau_{ew}}-\frac{N_{W}}{\tau_{wr}}$$
(2)$$\frac{\partial h(z,t)}{\partial t}=\frac{N_{W}(1-h)}{N_{Q}\tau_{we}}+\frac{f(1-h)}{\tau_{ge}}-\frac{h(1-f)}{\tau_{eg}}-\frac{h}{\tau_{ew}}-\frac{h^{2}}{\tau_{er}}$$
(3)$$\frac{\partial f(z,t)}{\partial t}=\frac{h(1-f)}{\tau_{eg}}-\frac{f(1-h)}{\tau_{ge}}-\frac{f^{2}}{\tau_{gr}}-\frac{v_{g}g_{\max}}{N_{Q}}(2f-1)(S^{+}(z,t)+S^{-}(z,t))$$
where the maximum modal gain is denoted by g_max_ [21]. h and f are the electron occupation probabilities in the ES and the ground state (GS), q is the electron charge, I is the injection current and N_w_ is the electron density in the wetting layer. In addition, τ_ew_ is the electron escape lifetime from the ES to the WL and τ_we_ is the electron relaxation lifetime from the WL to the ES. Spontaneous radiative lifetimes in the GS, ES and the WL are denoted by τ_gr_, τ_er_, and τ_wr_, respectively. τ_ge_ and τ_eg_ are the electron escape and relaxation lifetime from GS to the ES and vice versa. The group velocity of light and the volume density of QDs are denoted by v_g_ and N_Q_, respectively. In the case of OP-RSOAs, the first term of Equation (1) is removed. A term will also be added to the ES rate equation. Therefore, Equations (1) and (2) will be replaced with Equations (4) and (5), respectively. In other words, Equations (3)–(5) are the rate equations in the case of optical pumping.
(4)$$\frac{\partial N_{W}(z,t)}{\partial t}=-\frac{N_{W}(1-h)}{\tau_{we}}+\frac{N_{Q}h}{\tau_{ew}}-\frac{N_{W}}{\tau_{wr}}$$
(5)$$\frac{\partial h(z,t)}{\partial t}=\frac{N_{W}(1-h)}{N_{Q}\tau_{we}}+\frac{f(1-h)}{\tau_{ge}}-\frac{h(1-f)}{\tau_{eg}}-\frac{h}{\tau_{ew}}-\frac{h^{2}}{\tau_{er}}+\frac{v_{g}\alpha_{\max}}{N_{Q}}(1-2h)(S_{OP}{}^{+}(z,t)+S_{OP}{}^{-}(z,t))$$
where the ES carrier dynamics are described by the term (1–2 h). In other words, 2 h < 1 and 2 h > 1 stand for absorption and optical gain, respectively. In addition, the maximum modal absorption coefficient is denoted by α_max_. h increases toward 0.5 with the optical pump power due to the last term in Equation (5). When h equals 0.5, this term vanishes (1–2 h = 0). Therefore, when the value *h* is decreased temporarily, the term (1–2 h) becomes positive. As a result, absorption occurs via optical pumping [21]. In addition to the carrier rate equations, the signal and pump propagation equations need to be solved simultaneously to derive the desired results [2].
(6)$$\frac{\partial S^{+}(z,t)}{\partial z}+\frac{1}{v_{g}}\frac{\partial S^{+}(z,t)}{\partial t}=(g_{\max}(2f-1)-\alpha_{\mathrm{int}})S^{+}(z,t)$$
(7)$$-\frac{\partial S^{-}(z,t)}{\partial z}+\frac{1}{v_{g}}\frac{\partial S^{-}(z,t)}{\partial t}=(g_{\max}(2f-1)-\alpha_{\mathrm{int}})S^{-}(z,t)$$
(8)$$\frac{\partial S_{OP}{}^{+}(z,t)}{\partial z}+\frac{1}{v_{g}}\frac{\partial S_{OP}{}^{+}(z,t)}{\partial t}=(-\alpha_{\max}(1-2h)-\alpha_{\mathrm{int}})S_{OP}{}^{+}(z,t)$$
(9)$$-\frac{\partial S_{OP}{}^{-}(z,t)}{\partial z}+\frac{1}{v_{g}}\frac{\partial S_{OP}{}^{-}(z,t)}{\partial t}=(-\alpha_{\max}(1-2h)-\alpha_{\mathrm{int}})S_{OP}{}^{-}(z,t)$$
where S^+^, S^−^, S_OP_^+^, and S_OP_^−^ denote the forward and backward signal and pump. The material absorption coefficient is indicated by α_int_. The modal absorption coefficient of the pump is defined as α_abs_ = −α_int_ − α_max_ (1–2 h). The derived results are used to investigate the CW operation mode and the steady state. In the next section, possible comparable parameters between optically and electrically pumped QD-SOAs are investigated. These include optical gain, transparency power, transparency current, and maximum output densities. Dynamic properties are also investigated, such as gain recovery time. One group of QD is selected for simplicity.

### 2.2. Dynamic Algorithm

Coupled differential rate and signal propagation equations must be solved to investigate the performance of QD-RSOAs. Several algorithms have been suggested to solve these equations [22,23,24]. In this paper, these equations are solved using the time–space discrete method to evaluate the occupation probabilities and signal powers at each point. Instead of a retarded time frame, these equations are presented in a definite time frame [2,25]. Although signal and pump propagation equations are derived in a retarded time frame, the final results are expressed in an absolute time frame. In other words, the final results are shifted after being solved. By using this method, the effects of counter-propagation pulses are also considered. Euler approximations are used to calculate time and spatial derivatives [22]. The signal and pump pulses are calculated in each step. Time steps are denoted by ∆t. As a result, the spatial step must be v_g_∆t for our results to converge. Several numbers are tested to find an appropriate step size. Sufficient stability and accuracy are derived using ∆t = 50 fs. In addition, signals are delayed by the amount ∆t in each spatial step. Static simulation results are used to start the dynamic simulations [26,27]. After the steady-state is reached, the input signal is inserted at z = 0. This signal has a time-dependent Gaussian profile. First, carrier densities, signal power, and pump power are derived at the following spatial step. Then, the signals propagate through the device. At this step, the effect of backward signals is not considered. At the ending facet of the RSOA, the signals are reflected and absorbed by the mirror. The amount of the reflected signal is specified by the mirror’s properties. In the ideal case (R = 1), all the signal reflects through the device without being absorbed. When the signals are propagating backward, the forward signals are kept constant. This iteration is repeated until we reach the desired tolerance (1%). In this way, the effects of counter-propagating pulses are also considered. This process is demonstrated in Figure 2.

## 3. Results and Discussion

### 3.1. Continous Wave Operation Mode

After solving the coupled differential rate and signal propagation equations, the unsaturated occupation probabilities for both OP-RSOA and EP-RSOA can be obtained for all the derived eigenstates. These eigenstates include the GS, ES, and WL. The total unsaturated material gain can be calculated as g_us_ = g_max_ (2f_us_−1) − α_int_. In this equation, the GS’s unsaturated occupation probability is denoted by f_us_. Amplification is experienced for the injected input signal at z = 0, in both forward and backward directions. However, if the amplifier’s length exceeds a certain amount, there will be no gain for the input signal. This happens because, for z > L_max_, material loss equals material gain. Therefore, there is a maximum output intensity in the structure S_max_. At this point, the occupation probabilities are extracted: N_W_ (z:L_max_) = N_wmax_, f (z:L_max_) = f_max_, h (z:L_max_) = h_max_ [28]. The maximum output density can be obtained as:(10)$$S_{\max}=\frac{g_{m}}{\alpha_{\mathrm{int}}}S_{sat}$$

The output saturation density is given by:(11)$$S_{sat}=\frac{N_{Q}}{v_{g}g_{\max}\tau_{gr}}$$

Additionally, g_m_ for both the optical and electrical pumping methods are calculated by:(12)$$g_{m}^{EP}=g_{\max}\tau_{gr}\left(\frac{J}{N_{Q}eL_{w}}-\frac{Nw_{\max}}{N_{Q}\tau_{wr}}-\frac{h_{\max}^{2}}{\tau_{er}}-\frac{f_{\max}^{2}}{\tau_{gr}}\right)$$
(13)$$g_{m}^{OP}=g_{\max}\tau_{gr}\left(\frac{\alpha_{\max}v_{g}}{N_{Q}}(1-2h_{\max})S_{OP}-\frac{Nw_{\max}}{N_{Q}\tau_{wr}}-\frac{h_{\max}^{2}}{\tau_{er}}-\frac{f_{\max}^{2}}{\tau_{gr}}\right)$$

Therefore, the output density is governed by the term α_max_v_g_ (1–2 h_max_) S_OP_, in the case of the OP-RSOA, whereas in EP-RSOA, the bias current controls it. As mentioned before, the absorption of the optical pump is managed by the term (1–2 h_max_); thus, this term is related to S_OP_. In addition, the optical gain and the power dissipation are directly influenced by the transparency current. This parameter can be used to optimize the performance of the device. In the case of the EP-RSOA, the transparency current is derived when g^EP^_m_ = 0. In addition, the power in which g^OP^_m_ equals zero is the optical transparency power. They can be obtained from:(14)$$J_{tr}=L_{w}N_{Q}e\left(\frac{Nw_{\max}}{N_{Q}\tau_{wr}}+\frac{h_{\max}^{2}}{\tau_{er}}+\frac{f_{\max}^{2}}{\tau_{gr}}\right)$$
(15)$$S_{tr}=\frac{N_{Q}}{\alpha_{\max}v_{g}(1-2h_{\max})}\left(\frac{Nw_{\max}}{N_{Q}\tau_{wr}}+\frac{h_{\max}^{2}}{\tau_{er}}+\frac{f_{\max}^{2}}{\tau_{gr}}\right)$$

Therefore, the maximum output densities are defined in terms of the optical transparency power, optical pump, transparency current, and bias current.
(16)$$S_{\max}^{OP}=\frac{\alpha_{\max}}{\alpha_{\mathrm{int}}}(1-2h_{\max})(S_{P}-S_{tr})$$
(17)$$S_{\max}^{EP}=\frac{1}{L_{w}ev_{g}\alpha_{\mathrm{int}}}(J-J_{tr})$$

In both EP and OP cases, bias current and optical pump increase the maximum output density linearly. Additionally, the transparency power affects the maximum output density. On the other hand, the slope of the output density is increased because of the term α_max_(1–2 h_max_) in OP-SOAs. In addition, since S_tr_ decreases when α_max_ is increased, it can be used as an enhancing factor. It has been proven that, in both SOAs, the optical gain depends on h_max_, f_max_, S_max_, S_in_, h_us_, S_sat_, S_out_, and f_us_ [28]. The optical gain of the QD-RSOA is depicted in Figure 3 for the Continuous Wave (CW) operation mode. The maximum RSOA length can be determined from this figure. This length is dependent on the optical pump or bias current and the unsaturated gain. For bias currents of above 20 mA and pump powers of above 20 mW, the optical gain saturates lengths exceeding 8 mm for both the OP and EP devices. If the bias current or the optical pump is not large enough, this length will be lower. This is due to the lack of inserted carriers into the structure. If the length is larger than L_m_, the material loss will dominate the gain. Therefore, the device will act as an attenuator instead of an amplifier. This effect can be seen in Figure 3 for both EP and OP with a bias current of 10 mA and an optical pump power of 10 mW (simulation parameters are given in Table 1).

Therefore, the maximum output densities are defined in terms of the optical transparency power, optical pump, transparency current, and bias current. As is evident in Figure 4, less transparency currents and power are observed for increased values of g_max_. This is because the transparency value of the GS is inversely related to g_max_. For g_max_ values higher than 1700 m^−1^, the transparency current stabilizes with a bias current of 50 mA and a RSOA length of 8 mm. In contrast, in the case of the OP-RSOA, for values higher than 1000 m^−1^, with the optical pump of 80 mW and the same length, the transparency power is stabilized. In addition, because of scattering losses and carrier-dependent absorption, in this case, lower population probability is yielded in higher ground state modal gain. This effect generates lower transparency power [29]. Figure 4 illustrates the normalized transparency power and current as a function of the ground state modal gain. The OP method functions better, especially at low GS modal gains. It can be used expeditiously when a high gain is not needed, such as in signal processing applications. The transparency power in this method compared to the transparency current in the EP method is as low as ~0.7/0.1 at g_max_= 1500 m^−1^. At g_max_= 1900 m^−1^ this number rises to ~0.37/0.0025. This is highly desirable in high-gain QD-RSOAs. This outcome is depicted in Figure 5. It illustrates the normalized transparency power as a function of α_max_ for two distinct ground state modal gains. On the other hand, the transparency power is dependent on both the excited state modal gain and the population through the term α_max_(1–2 h_max_). However, for two of the designated g_max_ values, the variations of this term are almost indistinguishable, as depicted in Figure 5c. The evolutions of the ES and GS population probabilities are different when ES modal gain dominates GS modal gain. This is not the case in our interest. Even though lower transparency power is achieved in higher α_max_, the GS modal gain is a much more critical factor.

### 3.2. Pulse Operation Mode

Temporal state dynamics are fundamental in the pulse mode operation. Therefore, the changes in state population probabilities due to optical pumping are investigated in this section. Some studies have already examined temporal properties in QD-RSOAs with electrical pumping [3,30]. However, few studies have examined these properties in QD-RSOAs with optical pumping. Thus, our primary focus is on OP-RSOAs. The following parameters were used to derive our results [18,31,32]:

**Table 1 nanomaterials-12-02143-t001:** Simulation parameters.

Symbol	Value	Description
L	2 mm	RSOA length
H	0.25 µm	Height of the RSOA
W	4 µm	Width of the RSOA
g_max_	1400 m^−1^	Maximum modal gain
α_max_	1000 m^−1^	The maximum modal absorption coefficient
α_int_	200 m^−1^	Material absorption coefficient
τ_wr_	0.2 ns	Recombination lifetime for WL
τ_gr_	0.4 ns	Recombination lifetime for GS
τ_we_	3 ps	Relaxation lifetime from WL to ES
τ_ew_	1 ns	Escape lifetime from ES to WL
τ_eg_	0.16 ps	Relaxation lifetime from ES to GS
τ_eg_	1.2 ps	Escape lifetime from GS to ES

The ground state carriers are recombined radiatively via stimulated emission when a single pulse is inserted into the structure. Carriers in the excited state refill the GS, acting as a carrier reservoir. This enables ultrafast gain recovery time. When there is an electrical pump, carriers captured from the wetting layer refill the ES after a few picoseconds. Fast Auger-assisted relaxation is the reason for this process. This phenomenon happens because of the high carrier density in the WL. This cycle continues until the signal completely passes the cavity or the injected current is removed from the structure. Therefore, gain recovery time is affected by the injected current density. However, when there is an optical pump, the ES is refilled via carrier absorption. The optical pumping is adjusted to make this happen. The carriers’ relaxation from the WL also makes a minor contribution to this refilling process. This is the result of the recaptured carriers from the excited state to the wetting layer. This process is modeled via the relaxation lifetime constant (τ_ew_). Figure 6 and Figure 7 illustrate the time evolution of the electron occupation probability in the ground state and excited state. Electron concentration in the wetting layer is also depicted in these figures. These parameters are presented as a function of input pulse power and pump power. As illustrated in Figure 6, in higher optical pumping powers, the parameters reach their steady state faster, and the recovery time is also lower. This is because in higher pump power, more carriers are absorbed. This makes the refilling of the ES faster. In addition, in higher input powers, the number of carriers recombined radiatively is increased. These recombined carriers need to be refilled from ES. As a result, the recovery time is greater in higher input powers, as is depicted in Figure 7. Another factor that may need explanation is that, unlike conventional traveling SOAs in RSOAs, there are two recovery areas. This is because, in RSOA, both the input and output facets are the same (at z = 0). Therefore, the first one is because the input signal is inserted at t = 0. The next one is because the backward signal exits the device at t ≈ 45 ps. In other words, it takes about 45 ps for the signal to move twice across the cavity. The second drop is more significant because the signal is more amplified at the exiting point than the entering point. In EP-RSOAs, the GS and the ES recover faster when there is a single input pulse. This is because the maximum population probability in the excited state is about 0.5 in OP-RSOAs. In other words, the rate of carrier relaxation from the excited state to the ground state in the EP method is almost twice the OP method. In addition, the ES recovers more quickly because of the high carrier population in the WL in EP-RSOAs.

In the suggested model, QDs come with a two-dimensional wetting layer (WL). However, the recovery times of GS and the ES are entirely affected by the slow recovery of the WL when an input pulse train is inserted into the structure. As an example, a 5 Tbs^−1^ input pulse train has been inserted into the structure, as depicted in Figure 8 and Figure 9. After the first pulse, the GS is almost completely recovered. However, after the rapid decline in modulation depth, the GS population probability is stabilized at around 0.01. The GS population probability is not strongly affected in the transient mode because of the incomplete recovery of QD. Therefore, the gain modulation depth is not affected significantly. However, in the case of the OP, the modulation depth for the leading pulse is almost twice the stabilized modulation of the ground state population probability. It reaches about 0.05, making an open eye pattern. As a result, the dynamical properties of QD-RSOAs are significantly improved in the OP method. Figure 10 illustrates the WL dynamics for a 5 Tbs^−1^ input train pulse for the EP and OP methods [12].

Figure 11 depicts the gain versus input power in OP-RSOAs with distinct pump powers. The saturation power can be extracted from this figure. For example, the saturation power with 10 mW pump power is about 18 dBm. The saturation power in QD-RSOAs is governed by several parameters, such as electron capture rate into the quantum dot and the number of quantum dot layers. The modal gain is enhanced by increasing these layers. However, to achieve the same level of output saturation power, a higher optical pump or bias current density is needed. In this case, the same inversion level is obtained, resulting in the identical output saturation power. In addition, the minimum capture rate of electrons from the wetting layer to the excited state limits the maximum saturation power in EP-RSOAs. On the other hand, in the OP case, it is determined by the relaxation rate from ES to GS. In this region, the quasi-equilibrium between the reservoir electron states and the quantum dot falls apart. This results in intense spectral hole burning. Therefore, the maximum saturation power is inversely relevant to the capture time [12].

Figure 12a demonstrates gain versus input power for a QD-RSOA with distinct reflections. For low input powers, the gain value is almost constant. However, when the input power increases, the value of the gain drops smoothly. In this region, the device is saturated. Moreover, the GS carriers are depleted, and the pump power can no longer provide sufficient carriers to refill the GS. As is evident, when the reflection is increased, higher gains are obtained. Output power is also illustrated in Figure 12b. The plot is linear for low input powers; this is the wanted region for amplifiers. However, as we increase the input power, the slope decreases. Finally, QD-RSOAs are capable of providing higher gains in the same length compared to conventional QD-SOAs. This is because in their reflective configuration, the signal passes the cavity twice.

## 4. Conclusions

A new scheme for QD-RSOAs with optical pumping was suggested in this paper. Numerical solutions of rate and signal propagation equations were used to study the dynamical properties, such as the recovery process of the gain. On the other hand, to investigate CW operation mode and steady-state characteristics, analytical solutions were exploited. First, some properties such as the transparency power, transparency current, optical gain, and gain recovery time were analyzed in both OP and EP RSOAs. By studying the gain saturation characteristics in CW operation mode, the effective length of the device was determined. In addition, it was proven that the normalized transparency power in OP-RSOAs can be much lower than the normalized transparency current in EP-RSOAs at both low and high GS modal gains. Moreover, by studying the recovery processes at the ES, GS, and WL, it was proven that dynamical properties can be significantly improved in the optical pumping scheme. It was also demonstrated that optical pumping makes pattern-free operation possible at extremely high speeds (Tb/s). As a result, OP QD-RSOAs are impressive candidates as all-optical signal processors, all-optical logic gates, and WDM-PONs. Despite all the mentioned advantages, optical pumping has few drawbacks compared to EP-RSOAs. The small-signal gain and the maximum GS population probability are lower in OP-RSOAs. Although there is no need for a current source in the optical pumping scheme, another laser is required as an optical pump. This complicates the setup of the device. However, this pump can be integrated into the whole structure, making an all-optical fast device.

## Figures and Tables

**Figure 1 nanomaterials-12-02143-f001:**
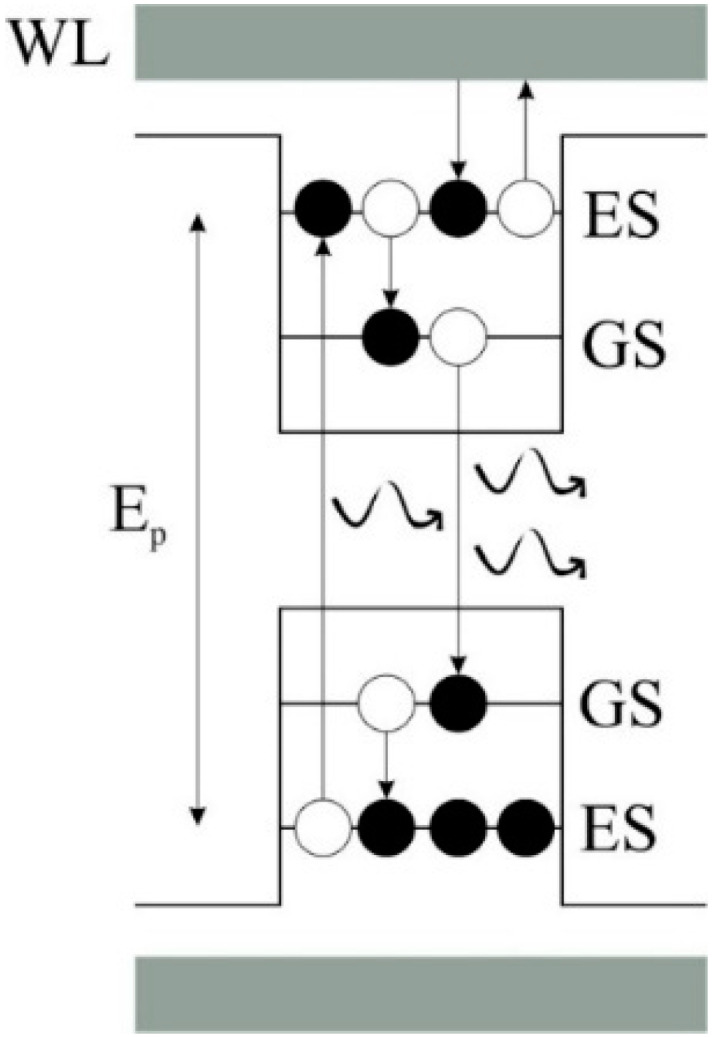
Energy band diagram for the proposed QD structure. This figure shows possible transitions between states that are considered in the rate equations.

**Figure 2 nanomaterials-12-02143-f002:**
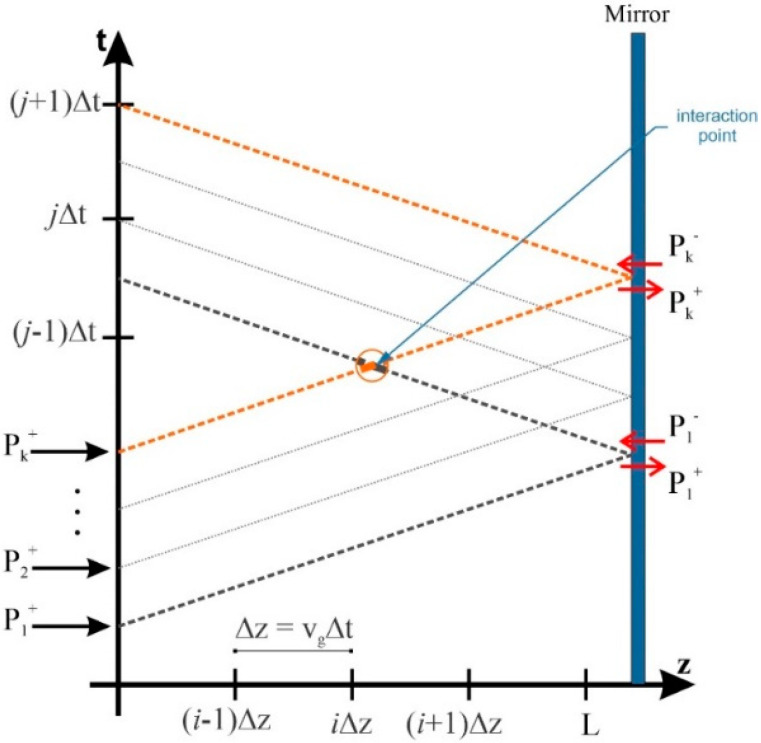
Backward and forward signals in the discrete time–space matrix.

**Figure 3 nanomaterials-12-02143-f003:**
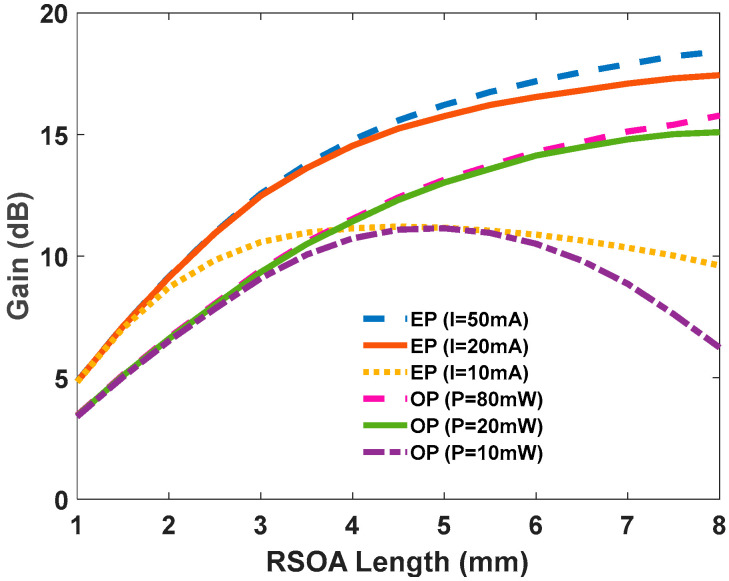
Optical gain of OP and EP QD-RSOAs versus the RSOA’s length for various pump powers and current densities.

**Figure 4 nanomaterials-12-02143-f004:**
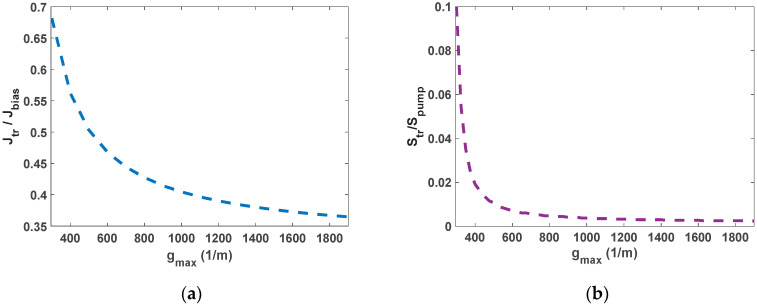
(**a**) Normalized transparency current for the EP QD-RSOA. (**b**) Normalized transparency power for the OP QD-RSOA.

**Figure 5 nanomaterials-12-02143-f005:**
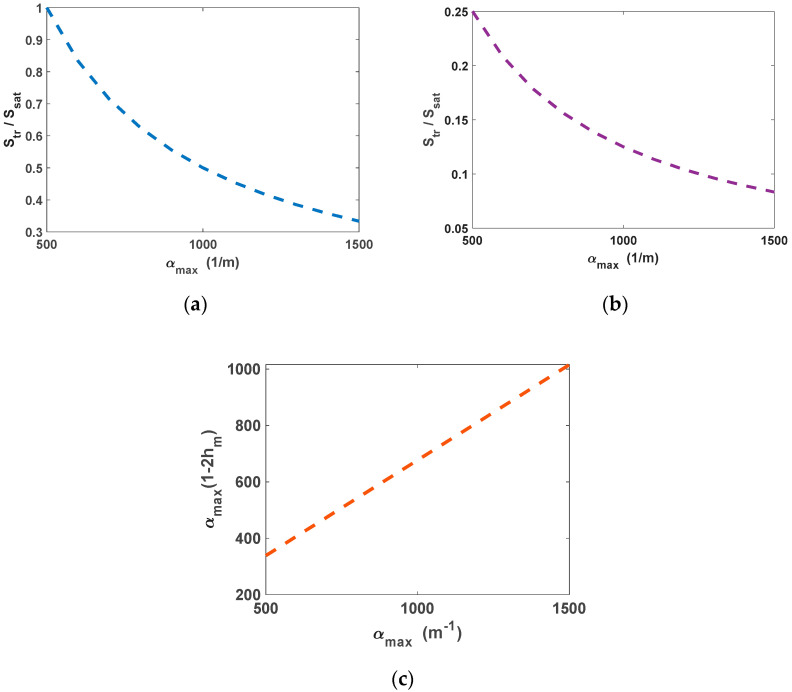
(**a**) Normalized transparency power versus ES modal gain for g_max_ = 1500 m^−1^. (**b**) Normalized transparency power versus ES modal gain for g_max_ = 1900 m^−1^. (**c**) The term α_max_ (1–2 h_m_) versus ES modal gain.

**Figure 6 nanomaterials-12-02143-f006:**
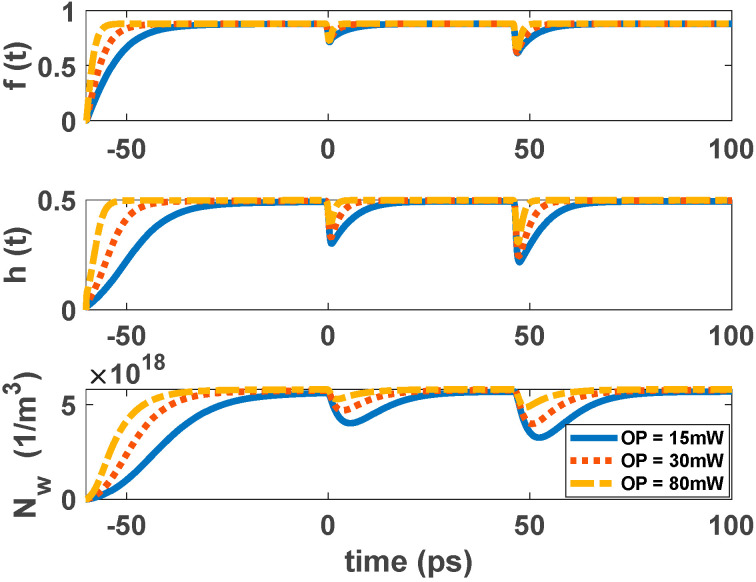
Occupation probability in the GS, ES, and electron concentration in WL as a function of time for different pump powers. Input signal power is 100 mW.

**Figure 7 nanomaterials-12-02143-f007:**
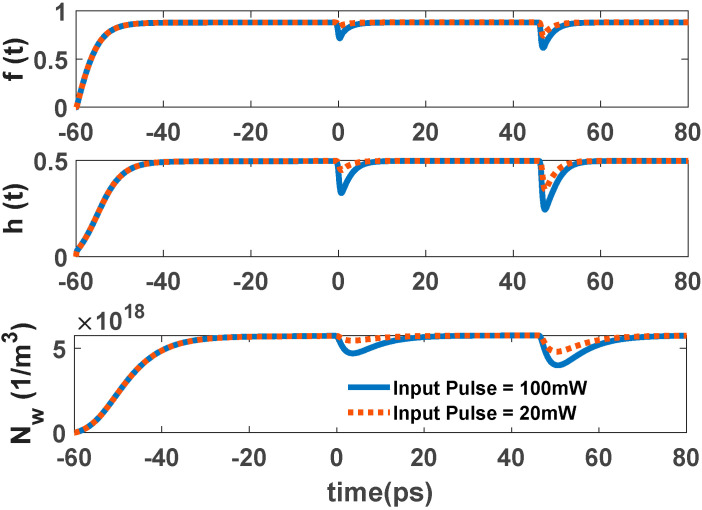
Occupation probability in the GS, ES, and electron concentration in WL as a function of time for different input pulse power. Optical pump power is 30 mW.

**Figure 8 nanomaterials-12-02143-f008:**
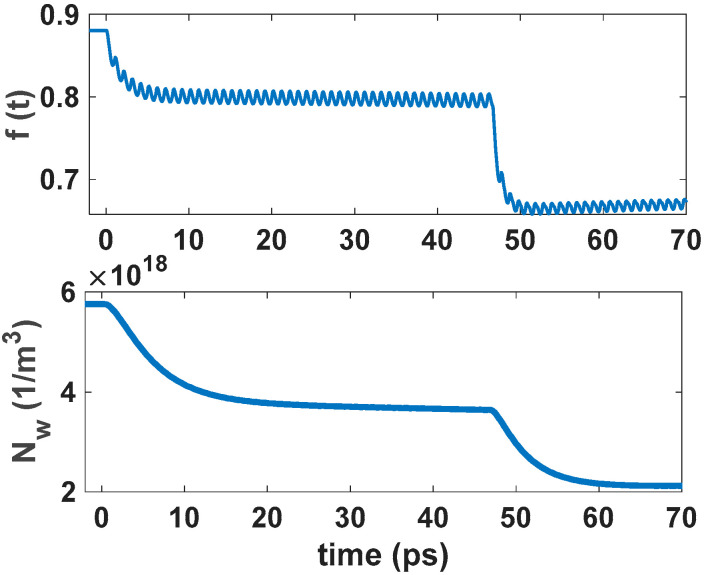
The GS and WL dynamics for a 5 Tb s^−1^ input bit sequence in the OP method. The input signal power is 20 mW.

**Figure 9 nanomaterials-12-02143-f009:**
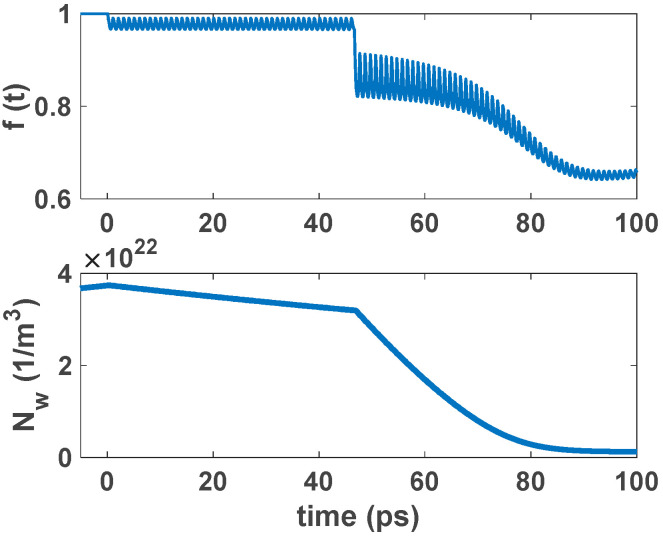
The GS and WL dynamics for a 5 Tb s^−1^ input bit sequence in the EP method. The input signal power is 20 mW.

**Figure 10 nanomaterials-12-02143-f010:**
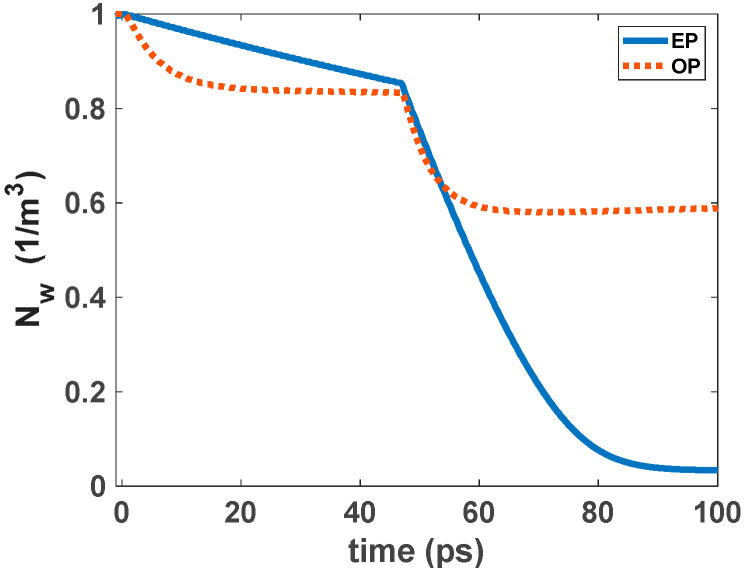
Normalized WL dynamics for both EP and OP methods. The plots are normalized to their maximum value.

**Figure 11 nanomaterials-12-02143-f011:**
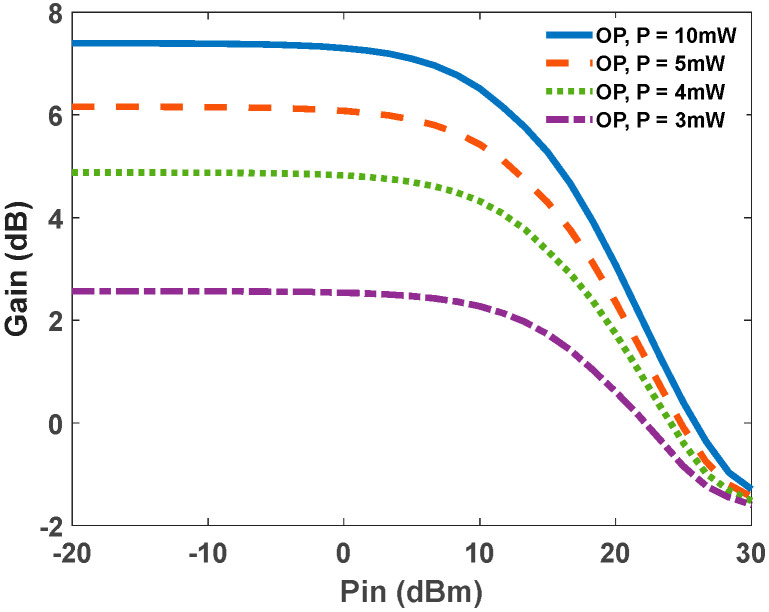
Gain versus input power for 4 distinct pump powers.

**Figure 12 nanomaterials-12-02143-f012:**
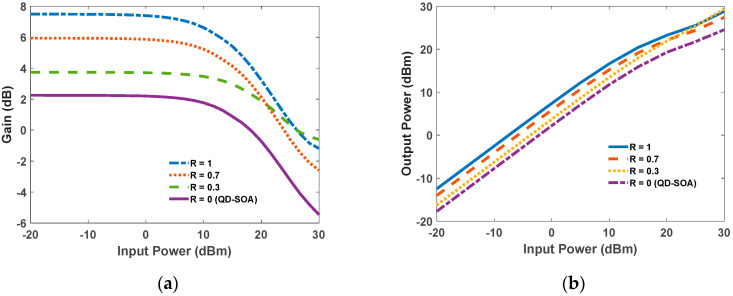
(**a**) Gain versus input power for different values of R. (**b**) Output power versus input power for different values of R.

## Data Availability

Not applicable.

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
