# Peer review of "Quantum Dot Reflective Semiconductor Optical Amplifiers: Optical Pumping Compared with Electrical Pumping"

_nanomaterials, 2022, doi:10.3390/nano12132143_

Round 1

Reviewer 1 Report

In the present work, the authors theoretically study a semiconductor optical amplifier made by quantum dots and working in reflection configuration. They compare the performances with optical and electric pump. They discuss the cw and pulsed operation and application to all-optical processing.

A few point must be clarified:

1.       It is not clear where the optical pump rate is in the rate equation. Is it represented by the injection current? If yes how is possible to distinguish between the optical and electric pump rate?

2.       The QDs scheme in figure 3 is misleading. Indeed, it seems that 1 electron is promoted to the upper GS and two in the upper ES, while only two vacancies are left in the lower GS and ES. That is the QDs has negative net charge. Can the author better explain?

3.       Have the author take into account the optical pump wavelength and what is the effect of different wavelength on the SOA performances?

4.       It is not clear with the occupation probability in the GS and ES shown in figure 6 and 7 are growing or decreasing at the same time. Shouldn’t they have the opposite behavior?

5.       In figure 8 and 9 the authors compare the dynamic behavior with optical and electric pumping. While shorted time response is demonstrated for OP, it also to be noted that an optical pulse is generally shorted with respect to an electric pulse. Have the author normalized the results in figure 8 and 9 the time duration of optical and electrical pulse?

6.       How does it change the dynamic behavior (fig 6, 7, and 8) with respect to the optical pulse duration?

7.       What exactly the author means for “Ultrawide Band” in their title?

Author Response

Dear Editor

Enclosed is the revised version of our paper submitted for your consideration and publication in Nanomaterials.

We acknowledged all comments in the paper and in the following a short response to each question is presented.

Bests

Ali Rostami

Reviewer 2 Report

The authors have investigated the quantum dot reflective semiconductor optical amplifiers with an optical pump. The comparison has been carried out between the quantum dot reflective semiconductor optical amplifiers with an optical pump and the quantum dot reflective semiconductor optical amplifiers with an electrical pump. The manuscript is interesting and useful for the design of quantum dot reflective semiconductor optical amplifiers, and is acceptable to be published in Nanomaterials, provided the following issue can be addressed

1.     Specify the full names of all abbreviations for the first appearance. 

2.     Add a table to specify the physical parameters used in the considered optical amplifier

3.     Add some discussion regarding the potential application of the considered optical amplifier in wideband communication systems.

See e.g.

C Jin et al., Nonlinear coherent optical systems in the presence of equalization enhanced phase noise, IEEE Journal of Lightwave Technology, 2021.

FS Nahaei et al., Selective band amplification in ultra-broadband superimposed quantum dot reflective semiconductor optical amplifiers, Applied Optics, 2022.

Author Response

(The authors gave the same response as above.)

Round 2

Reviewer 1 Report

The authors have addressed all the raised points.